# The magnitude and time course of pre-saccadic foveal prediction depend on the conspicuity of the saccade target

Lisa M Kroell[1,2]*, Martin Rolfs[1,2,3,4]

[1]Department of Psychology, Humboldt-Universität zu Berlin, Berlin, Germany; [2]Berlin School of Mind and Brain, Humboldt-Universität zu Berlin, Berlin, Germany; [3]Exzellenzcluster Science of Intelligence, Technische Universität Berlin, Berlin, Germany; [4]Bernstein Center for Computational Neuroscience Berlin, Berlin, Germany

## eLife Assessment

This study reports **important** findings about pre-saccadic foveal prediction and the extent to which it is influenced by the visibility of the saccade target relative to its background. The research methodology and results make a **convincing** case that foveal congruency effects develop when salient local contrast variations at the saccade target location can be used to direct the eye movement. This work should be of broad interest to visual neuroscientists, as well as those interested in understanding perception in the context of eye movements and in modeling visually guided actions.

*For correspondence:
lisa.m.kroell@gmail.com

Competing interest: The authors declare that no competing interests exist.

**Abstract** We previously demonstrated that during the preparation of a saccadic eye movement, human observers anticipate defining features of the eye movement target in pre-saccadic foveal vision (Kroell and Rolfs, 2022). In this Research Advance, we show that the conspicuity of feature information at the saccade target location influences the magnitude and time course of foveal enhancement. Observers prepared a saccade to a peripheral orientation signal (the target) while monitoring the appearance of another orientation signal (the probe) in their pre-saccadic center of gaze. The foveal probe appeared in 50% of trials and either had the same orientation as the target (congruent) or a different orientation (incongruent). Crucially, we manipulated the opacity of the target against the 1/f noise background in four logarithmic steps (25–90%). An increase in opacity translated to an increase in luminance contrast and signal-to-noise ratio of orientation information within the target region. Foveal enhancement defined as the difference between hit rates to target-congruent and target-incongruent foveal probes increased with target opacity. Moreover, the time course of foveal enhancement showed an oscillatory pattern that was particularly pronounced at high target opacities. Reverse correlations furthermore suggest that at higher target opacities, false alarms were increasingly triggered by signal, i.e., by incidental orientation information in the foveal noise. Beyond providing new mechanistic insights, these findings are relevant for researchers planning to adapt our paradigm to study related questions. Presenting the saccade target at a high signal-to-noise ratio appears beneficial, as foveal congruency effects, especially when time-resolved, are most robustly detectable.

## Introduction

We previously demonstrated that during the preparation of a saccadic eye movement to a peripheral visual field location, defining features of the eye movement target are predictively enhanced in pre-saccadic foveal vision (**Kroell and Rolfs, 2022**). In this Research Advance, we manipulated the

conspicuity of the saccade target in a parametric fashion and, based on a large number of trials, described the influence of this manipulation on the magnitude and temporal development of foveal prediction.

In our original paradigm, we presented a rapid sequence of 1/f noise images filling the entire screen (see *Hanning, 2022*; *Hanning and Deubel, 2022*). Human observers maintained fixation in the screen center until an orientation signal, generated by filtering the luminance content at the desired display location to either a –45° or 45° tilt, appeared 10 degrees of visual angle (dva) to the left or the right of the screen center. Observers executed an eye movement to the peripheral orientation signal (i.e. the target) as soon as they detected it. During saccade preparation, a second orientation signal (the foveal probe) could appear in observers' current, pre-saccadic center of gaze on 50% of trials. If presented, the probe had either the same orientation as the target (congruent) or a different orientation (incongruent). After executing the saccade, observers reported if they had perceived the probe in the screen center before moving their eyes and, if so, which orientation they had perceived.

Starting 175 ms before saccade onset, observers' hit rates (HRs) for foveal probes with target-congruent orientation significantly exceeded incongruent HRs. This congruency effect was spatially confined to the center of gaze and its immediate vicinity and more pronounced during saccade preparation than during passive fixation. Based on these findings, we proposed that feedback connections from higher-order visual areas enhance relevant saccade target features in foveal retinotopic cortex (see also *Williams et al., 2008*; *Chambers et al., 2013*; *Fan et al., 2016*; *Weldon et al., 2016*; *Yu and Shim, 2016*; *Weldon et al., 2020*). Internal orientation predictions subsequently combine with feedforward foveal input to early visual cortex and facilitate the detection of target-congruent feature information in the center of gaze.

In all previous experiments, we presented the saccade target at an opacity of 60% against the background noise. Importantly, the choice of a certain target opacity in our paradigm dictates the luminance contrast, as well as the signal-to-noise ratio (SNR) of orientation information within the saccade target region. We had chosen a medium target opacity in previous experiments since we presumed that feature information may need to be relevant for the movement task to be predicted foveally. In other words, while high target opacities would likely have generated a salient and immediate target pop-out, continuous peripheral sampling of orientation information (see *Ludwig et al., 2014*) may have been necessary to program the eye movement at a medium opacity. On the other hand, a highly conspicuous saccade target may not only facilitate the oculomotor task and free up resources for perceptual detection but could benefit foveal prediction more directly: the higher the SNR of peripheral orientation information, the less noisy the foveally predicted signal may be. Moreover, the timing of foveal feedback effects during fixation varied considerably across previously published investigations (from 117 ms in *Weldon et al., 2016*; *Weldon et al., 2020*, to 550 ms in *Fan et al., 2016*). While this variation has been ascribed to a flexible adjustment of the feedback mechanism to current task demands (*Fan et al., 2016*), low-level stimulus features may play an equally important role, at least in our investigations. In particular, neuronal response latencies decrease systematically as the contrast of visual input increases. While this phenomenon is reliably observed at varying stages of the visual processing hierarchy—such as the lateral geniculate nucleus (*Lee et al., 1981*), primary visual cortex (e.g. *Albrecht, 1995*; *Carandini and Heeger, 1994*; *Carandini et al., 1997*; *Carandini et al., 2002*), and anterior superior temporal sulcus (STSa; *Oram et al., 2003*; *van Rossum et al., 2008*)—influences of contrast on neuronal response latency are particularly pronounced in higher-order visual areas: a doubling of stimulus contrast has been shown to decrease the latency of V1 neurons by 8 ms, compared to a reduction of 33 ms in area STSa (*Oram et al., 2003*; *van Rossum et al., 2008*). Assuming that the peripheral target is processed in a bottom-up fashion until it reaches higher-order object processing areas, the time point at which peripheral signals are available for feedback should be dictated by the temporal dynamics of visual feedforward processing.

In this Research Advance, we describe the influence of saccade target opacity on pre-saccadic foveal congruency effects. Note that throughout the paper, we use the term 'opacity' when we refer to the experimental manipulation (a variation of the transparency, i.e. 1-opacity of the target patch against the background noise) and the term 'conspicuity' when we discuss our findings conceptually. We varied target opacity in four steps (equally spaced on a log scale: 25.0%, 38.3%, 58.7%, 90.0%) while leaving the remaining task parameters unchanged (*Figure 1*). An increase in target opacity from 25% to 90% corresponded to an increase in *Michelson, 1927* from 0.53 to 0.78 and an increase in

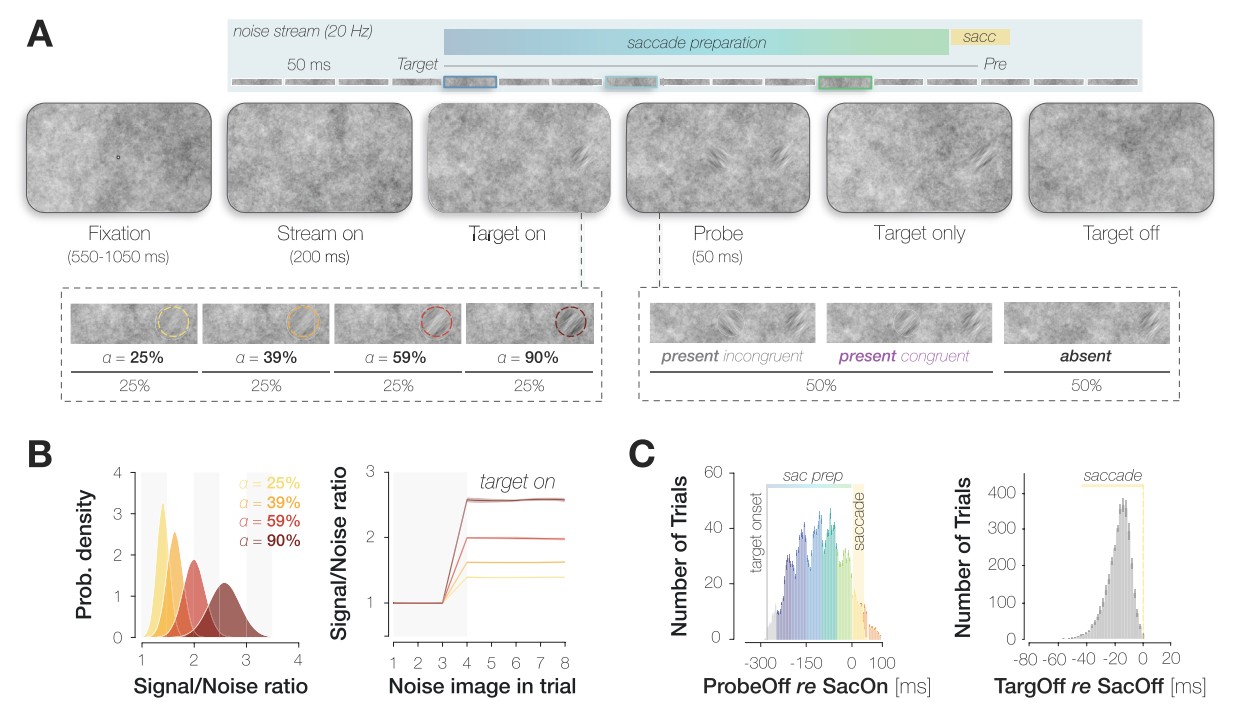

**Figure 1.** Summary of the paradigm. (**A**) Example trial procedure: the saccade target and foveal probe were embedded in full-screen noise images flickering at a frequency of 20 Hz (image duration of 50 ms). After 200 ms, the saccade target (an orientation-filtered patch; filtered to either –45° or +45°; 3 degrees of visual angle [dva] in diameter) appeared 10 dva to the left or the right of the screen center, cueing the eye movement. On 50% of trials, a probe (a second orientation-filtered patch; filtered to either –45° or +45°) appeared in the screen center at an early (top panel; highlighted element with dark blue outline), medium (light blue outline), or late (green outline) stage of saccade preparation. The foveal probe was presented for 50 ms and could be oriented either congruently or incongruently to the target. In contrast to our previous investigation, the saccade target was presented at one of four opacities (from 25% to 90%; in different blocks). (**B**) An increase in target opacity translates to an increase in signal-to-noise ratio (SNR; left panel). Within a single trial, the increase in SNR at the target location manifests from the fourth noise image on (i.e. after the target appears; right panel; error bands correspond to the standard deviation across all images). (**C**) Probe and target timing. Probe timing (left panel): histogram of time intervals between probe offset and saccade onset. Bar heights and error bars indicate the mean and standard error of the mean (SEM; n=9) across observers, respectively. On included trials, the probe appeared after target onset and therefore during saccade preparation ('sac prep'). Trials in which the probe disappeared more than 250 ms before saccade onset (light gray), during the saccade (yellow) or after saccade offset (orange) were excluded. The yellow background rectangle illustrates the median saccade duration. Target timing (right panel): histogram of time intervals between target offset and saccade offset. Bar heights and error bars indicate the mean and SEM (n=9) across observers, respectively. Unlike in the previous study, we removed the target upon saccade initiation on all trials.

the SNR of the filtered orientation from 1.40 to 2.58 (see Methods). The foveal probe was presented at the same opacity, which was individually adjusted for every session and observer across all target opacity conditions. The difference between congruent and incongruent HRs for foveal probes, i.e., enhancement, increased monotonically with target opacity. This finding endorses the assumption that predictive foveal processing can support perceptual continuity in natural visual environments, in which saccades are routinely directed toward the currently most conspicuous object in the scene (*'t Hart et al., 2013*). Moreover, especially when systematic variations of saccade latency were taken into account, enhancement required less and less time to develop as the opacity of the saccade target, and thus its contrast, increased. These findings not only provide new mechanistic insights but refine our method and are relevant for researchers planning to use or adapt our paradigm to study related questions.

## Results
### Hit and false alarm rates as a function of target opacity
Both congruent and incongruent HRs decreased with increasing target opacity (*Figure 2A*, left panel), arguably because attentional resources were increasingly drawn to the target the more salient it

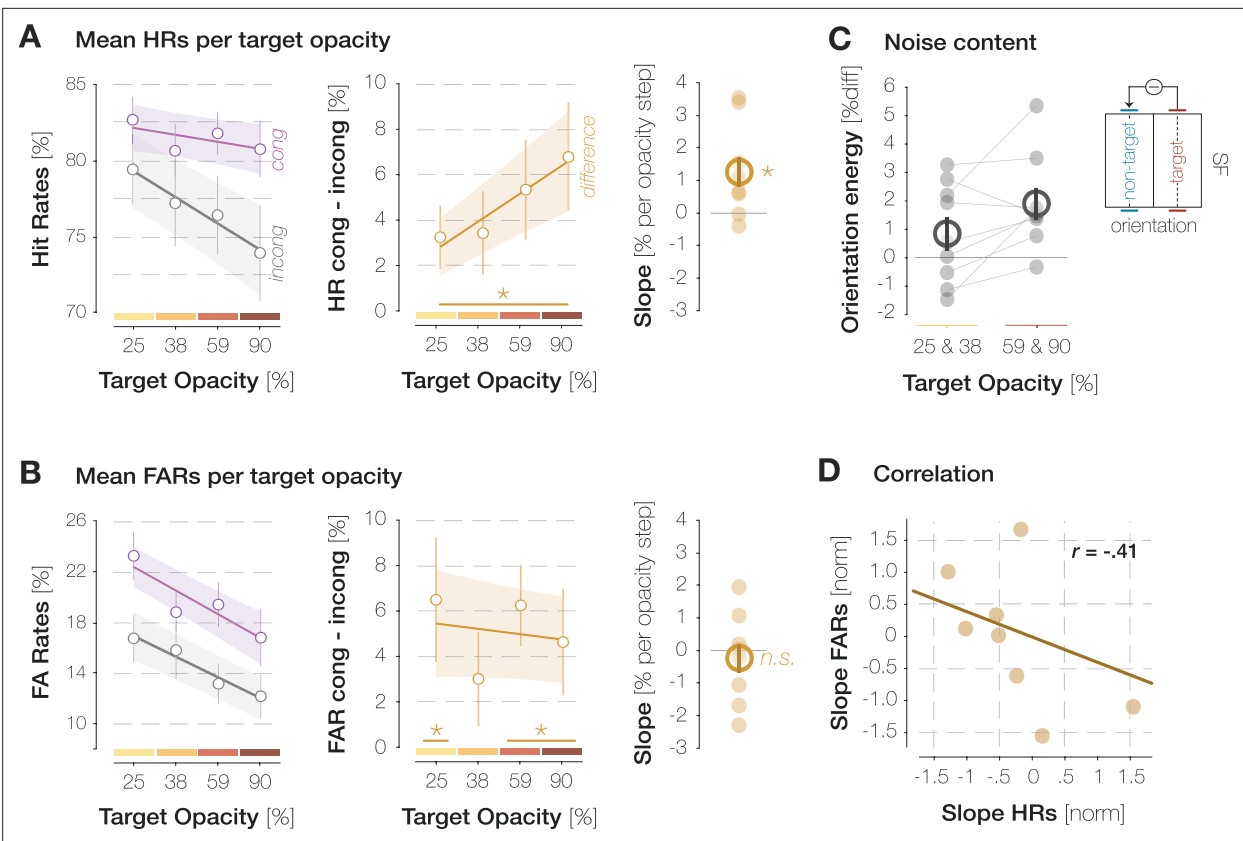

**Figure 2.** Influence of target opacity on hit rates (HRs) and false alarm rates (FARs) across all time points. (**A**) Influence of target opacity on congruent and incongruent HRs (purple and gray data points in the left panel), as well as their difference (orange data points, middle panel). Lines and error bands correspond to the fitted linear regression lines ± 2 standard errors of the mean (SEM). The slopes of the fitted regression line per observer (small circles) and their mean and SEM (big circle and error bar) are plotted in the right panel. Asterisks denote statistically significant comparisons (p<0.05; determined via bootstrapping; n=9 observers). (**B**) Influence of target opacity on congruent and incongruent FARs (left panel), as well as their difference (middle and right panels). All conventions are as in A. (**C**) Mean difference in filter energies around the target and non-target orientation for the lower two target opacities (left column) and the higher two target opacities (right panel). Lines connect the values of individual observers (small circles) in both conditions. Large circles and error bars denote the mean and SEM, respectively. (**D**) Pearson correlation between the normalized slope in HRs (A, right panel) and the normalized slope in FARs (B, right panel). Circles indicate individual observers.

became. This decrease in foveal HRs, however, was less pronounced for probes with target-congruent orientation (slope of the linear regression line: –0.46% [congruent] vs –1.73% [incongruent] per opacity step, p<0.001). In all four opacity conditions, observers detected foveal probes more readily if they exhibited the same orientation as the saccade target, resulting in significantly higher congruent as compared to incongruent HRs (HR$_{cong-incong}$=3.3%, 3.4%, 5.4–6.8%, ps<0.021; *Figure 2A*, middle panel). To fully characterize the impact of target opacity on enhancement in HRs, we performed a linear mixed-effects model in which we described the variance in enhancement (HR$_{cong}$–HR$_{incong}$) across all time points with a fixed effect of target opacity and a random intercept for observer (see Methods). Note that this model outperformed the simplest model including a fixed effect of target opacity only (ΔBIC = 11.6) and a more complex one involving a random intercept and random slope for observer (ΔBIC = 4.4). The fixed effect of target opacity reached significance, t(34) = 2.341, p=0.025. More specifically, as the opacity of the saccade target increased, the difference between HRs for target-congruent and target-incongruent foveal probes increased as well, resulting in a significantly positive slope of the fitted linear regression line (slope = 1.3% per opacity step, p<0.001; *Figure 2A*, right panel).

On probe-absent trials, observers reported perceiving the target orientation more often than the non-target orientation for the 25%, 59%, and 90% opacity conditions, resulting in significantly higher congruent as compared to incongruent false alarm rates (FARs; FAR$_{cong-incong}$=6.5%, 6.2, and 4.6%, ps<0.02; *Figure 2B*). We performed a linear mixed-effects model to describe the difference between

congruent and incongruent FARs with a fixed effect of target opacity and a random intercept for observer. Unlike for HRs, the fixed effect of target opacity did not affect the differences in FARs, t(34) = –0.44, p=0.661. The slope of the fitted linear regression line was statistically indistinguishable from zero (–0.2% per opacity step, p=0.713). Inspecting congruent and incongruent FARs instead of their difference revealed that, just like HRs, they both decreased with increasing target opacity. This decrease, however, was comparable for congruent and incongruent FARs (slopes: –1.9% vs –1.6% per opacity step, p=0.715).

In our paradigm, the entire screen was covered with pink noise. In consequence, the display region the probe could appear in was never void of signal but contained incidental orientation information even on probe-absent trials. Trials in which observers reported perceiving the probe when no stimulus was presented may therefore contain a mixture of two response types: (i) 'true' FAs, i.e., unsystematic or biased reports and (ii) 'true Hits', i.e., systematic reactions to incidental orientation information in the foveal noise region. As target opacity (and, presumably, the strength of the fed-back signal) increased, the proportion of 'true Hits' within the subset of congruent FAs may have increased while the proportion of the 'true FAs' decreased. The relative contribution of both response types may thus have varied considerably without affecting the mean congruent FAR. Indeed, two observations suggest that the contribution of unsystematic responses to FAs in general, and congruent FAs in particular, decreased as target opacity increased.

First, we described the visual properties of the foveal noise on FA trials by convolving a set of Gabor filters with different spatial frequencies (SFs) and orientations with the pixel content in a 3 dva region around the pre-saccadic center of gaze. We then subtracted filter energies on incongruent FA trials from filter energies on congruent FA trials (see Methods). If FAs relied partially on signal, this difference image should be characterized by high filter energies around the target orientation and low filter energies around the non-target orientation. Interestingly, the difference in filter energies around the target and non-target orientation was significantly more pronounced for the higher as compared to the lower two target opacities (p=0.008; *Figure 2C*). This observation suggests that the contribution of 'true Hits' to the pool of foveal FAs increased with target opacity, allowing systematic orientation effects to manifest in average noise properties.

Second, we related the dependence of HRs on target opacity to the dependence of FARs on target opacity across observers. Specifically, we correlated the z-standardized slope of the difference in HRs (*Figure 2A*, right panel) with the z-standardized slope of the difference in FARs (*Figure 2B*, right panel). This analysis yielded a moderate yet nonsignificant negative correlation between HR and FAR slopes (r=–0.41, p=0.278). In other words, observers who showed more enhancement in HRs as target opacity increased tended to show a smaller difference between congruent and incongruent FARs with increasing opacity. This correlation is based on a small number of observers (n=9) and relates two different measures which can be considered noisy estimates to begin with. It should thus be validated in future investigations involving a larger sample. Nonetheless, it suggests that higher target opacity levels increased observers' sensitivity for congruent foveal orientation signals.

After establishing the impact of target opacity on HRs and FARs across all time points, we aimed to characterize the influence of target opacity on the time course of foveal enhancement. Yet, we expected target opacity to influence saccade latencies in a systematic fashion, potentially complicating a direct comparison of time courses between opacity conditions. Three interdependent factors are likely to influence the time course of congruency effects in HRs: (1) the duration for which the target had been visible when the foveal probe appeared (target-locked time course), (2) the stage of saccade preparation at which the probe was presented (saccade-locked time course), and (3) the latency of a given eye movement which moderates the relation between (1) and (2). In consequence, a probe stimulus presented, for instance, 150 ms after the target will appear in later stages of saccade preparation as saccade latency decreases.

## Saccade parameters

Median saccade latencies decreased as the opacity and, therefore, the contrast of the saccade target increased (median mdn = 288.8, 279.7, 276.4, and 268.0 ms; *Figure 3A*). The slope of the linear regression line fitted to median latencies on an individual-observer level was significantly smaller than zero (mean m=–6.57 ms/opacity step, p<0.001). Target opacity did not affect the precision of saccadic timing: the slope of the regression line fitted to the standard deviation of latencies on an

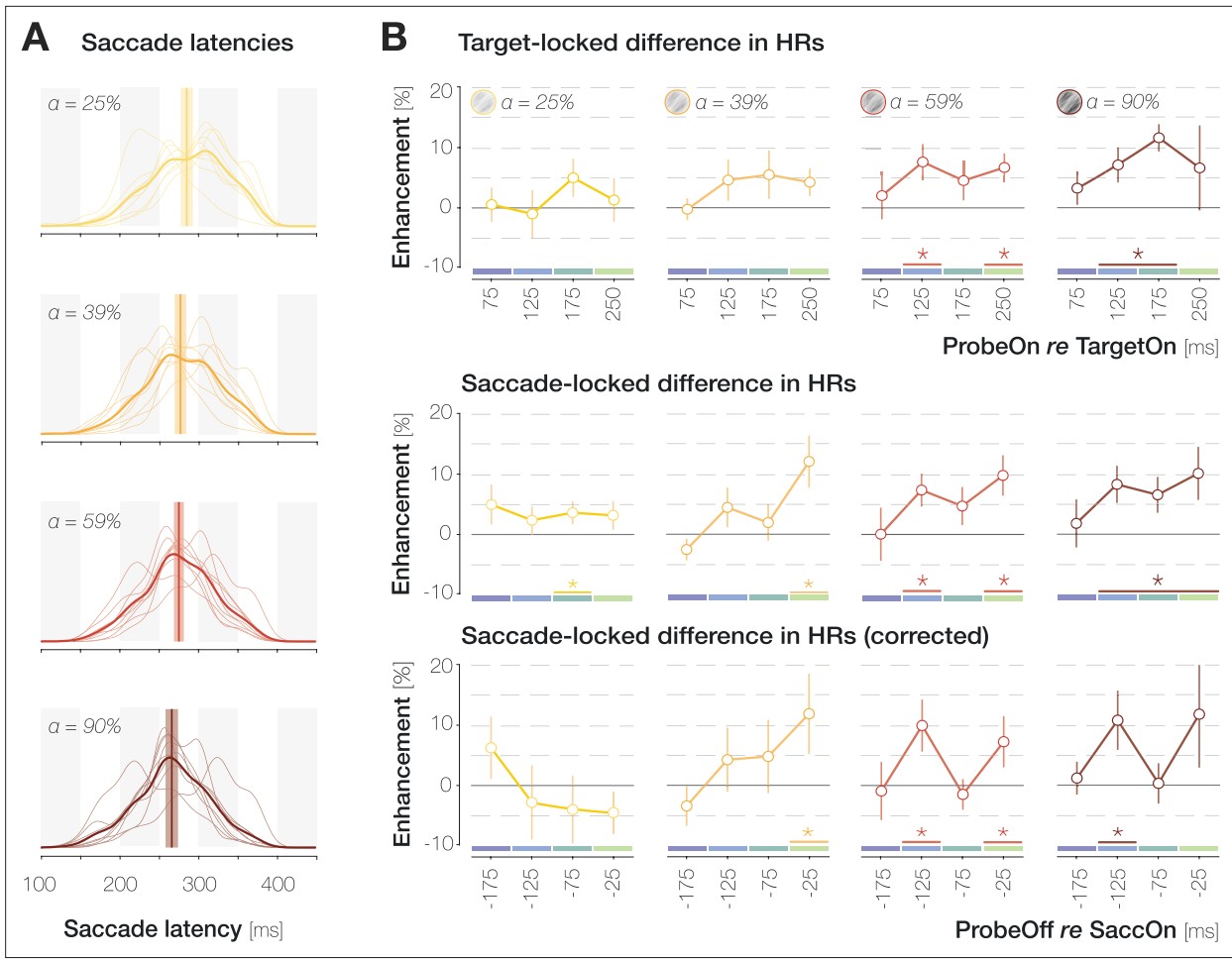

**Figure 3.** Time course of enhancement in hit rates (HRs) for different target opacity levels. (**A**) Probability density distributions of saccade latencies for increasing target opacity. Distributions with thin and thick lines represent individual-observer and mean probability densities, respectively. Vertical lines and shaded regions represent median latencies and standard errors of the mean (SEMs), respectively. (**B**) Target- (first row) and saccade- (second and third row) locked time course of enhancement. To obtain the corrected saccade-locked time course, the proportion of different target-locked bins in each saccade-locked bin was equalized across opacities to account for the systematic decrease in latencies with increasing opacity (see A). X-axis values indicate the center of 50 ms bins. Note that for saccade-locked time courses, the last bin contains probe onset times between 200 and 300 ms to allow for sufficient trial numbers. Across panels, error bars indicate SEMs. Asterisks denote significant differences between congruent and incongruent HRs (p≤0.05; determined via bootstrapping with 10,000 repetitions; n=9 observers).

individual-observer level was not significantly smaller than zero (m=–0.41 ms/opacity step, p=0.169). To fully characterize the influence of target opacity on saccade metrics, we computed bivariate Gaussian kernel densities of observers' saccade landing coordinates per opacity (***Appendix 1—figure 1***; Supplementary Materials). The center of mass fell inside the target region in all conditions, demonstrating that observers were able to execute accurate saccades even at the lowest opacity. Indeed, saccade amplitudes were uninfluenced by target opacity (mdn = 10.20, 10.31, 10.25, and 10.33 dva): the slope of the regression line fitted to median amplitudes on an individual-observer level was statistically indistinguishable from zero (m=0.03 dva/opacity step, p=0.153, ***Appendix 1—figure 1D, panel 2***). Likewise, saccadic error defined as the Euclidean distance between saccade landing coordinates and the center of the target stimulus did not vary with target opacity (mdn = 1.35, 1.35, 1.38, and 1.37 dva, p=0.171; ***Appendix 1—figure 1D, panel 3***). Saccadic peak velocities, on the other hand, decreased significantly with opacity (mdn = 416.7, 413.70, 410.17, 407.25 dva/s, p<0.001; ***Appendix 1—figure 1D, panel 4***). However, due to the small absolute scale of these variations, the main sequence defined as the relation between saccade amplitudes and peak velocities remained largely unaltered (***Appendix 1—figure 1C***).

## The influence of target opacity on the time course of enhancement

When probe onset was temporally aligned to the *onset of the peripheral target stimulus* (*Figure 3B*, first row), congruent and incongruent HRs did not differ significantly in any time bin for the lowest two opacities (all ps between 0.085 and 0.686; obtained with bootstrapping, see Methods). For targets with 59% opacity, congruent HRs significantly exceeded incongruent ones if the probe appeared 100–150 or 200–300 ms after the target (p=0.003 and p<0.001). For the highest target opacity, we observed a continuous enhancement window ranging from 100 to 200 ms after target onset (ps<0.003). We subsequently inspected the saccade-locked time course by temporally aligning the offset of the probe stimulus to the *onset of the eye movement*. The saccade-locked development of foveal enhancement roughly mirrors target-locked time courses (*Figure 3B*, second row). Yet, due to the systematic shortening of saccade latencies with target opacity (see *Figure 3A*), the same saccade-locked time bin contained trials with systematically different target-probe intervals for different target opacities. In the last pre-saccadic bin, for instance, the probe had appeared 211 ms after the target for the lowest opacity and 178 ms after the target for the highest opacity. To control for the influence of target-probe intervals on saccade-locked time courses, we determined the proportion of each target-probe offset in every pre-saccadic time bin. We then equalized these proportions across target opacities. To achieve this, we expressed the proportions in the 59% opacity condition (i.e. approximately the opacity used in our previous investigations) as multiples of the proportions in every remaining opacity condition. We subsequently weighted the saccade-locked enhancement values by these relative proportions (see *Rolfs and Carrasco, 2012*, for a similar approach). As a result, enhancement in each saccade-locked bin in *Figure 3C* is reconstructed to contain similar probe-target offsets across opacities.

We did not observe a significant difference between congruent and incongruent HRs in any pre-saccadic time bin for the lowest target opacity (all ps between 0.095 and 0.905). At an opacity of 39%, significant enhancement manifested during the 50 ms immediately preceding saccade onset, p=0.006. As target opacity increased further, enhancement was additionally observable in an earlier saccade preparation bin from 150 to 100 ms before eye movement onset, p=0.003. At the highest

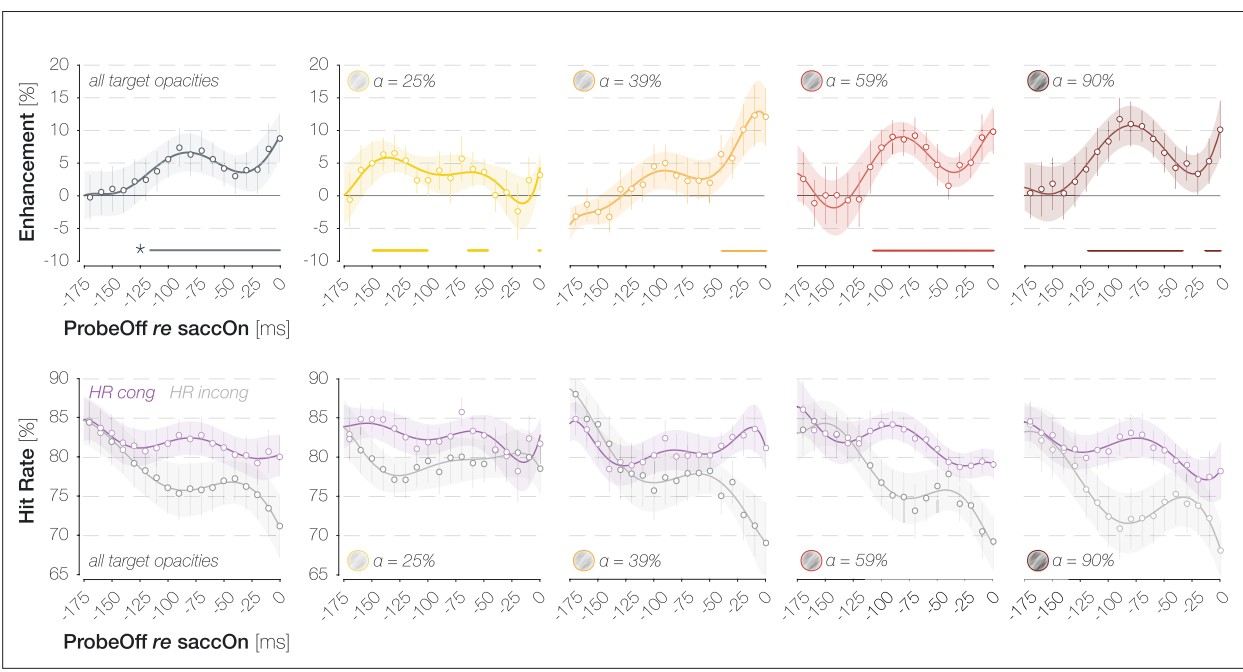

**Figure 4.** Continuous time course of foveal congruency effects. First row: continuous time course of enhancement in hit rates (HRs) (HR$_{cong}$–HR$_{incong}$) across target opacities (gray; plot 1) and separately for different target opacity levels (plots 2–5; yellow to dark red). X-axis values indicate the latest time point in a sliding boxcar window of 50 ms duration. Data points and error bars correspond to the mean ± 1 standard error of the mean (SEM) across observers. Lines and error bands correspond to mean sixth-order polynomial fits ±1 SEM. Horizontal lines above the x-axes denote significant enhancement (p≤0.05; determined via bootstrapping with 10,000 repetitions; n=9 observers). Second row: same plots as in the first row but separately for congruent (purple) and incongruent (gray) HRs.

target opacity, congruent HRs exceeded incongruent ones in the 150–100 ms bin exclusively, p=0.005. In short, the corrected saccade-locked time courses appear to suggest that foveal enhancement is observable in earlier stages of saccade preparation as target opacity increases.

Especially at higher target opacities, the temporal development of foveal enhancement appears to exhibit an oscillatory pattern. To inspect this incidental observation in a more temporally resolved fashion, we determined mean enhancement values in a boxcar window of 50 ms duration sliding along all saccade-locked probe offset time points (step size = 10 ms; x-axis values in *Figure 4* indicate the latest time point in a certain window). We then fitted sixth-order polynomials (with no constraints on parameters) to the resulting time courses and determined significant time points using bootstrapping (see Methods). The average foveal enhancement across target opacities reached significance starting 115 ms before saccade onset (gray curve in *Figure 4*; all ps<0.046). For every individual target opacity condition, we observed significant enhancement immediately before saccade onset, although only very briefly for the lowest opacity (−2 to 0 ms for 25%; −39 to 0 ms for 39%, −106 to 0 ms for 59%, and −13 to 0 ms for 90%; all ps<0.050; yellow to dark red curves in the first row of *Figure 4*). Especially for the higher two target opacities, we observed a local maximum preceding eye movement onset by approximately 80 ms. Interestingly, assuming a peak in enhancement in 80 ms intervals (i.e. at x-axis values of −80 and 0 ms in *Figure 4*) would correspond to an oscillation frequency of 12.5 Hz. In contrast to rapid feedforward processing, feedback signaling is associated with neural oscillations in the alpha and beta range (i.e. between 7 and 30 Hz; *Bastos et al., 2015*; *Jensen et al., 2015*; *van Kerkoerle et al., 2014*). We subsequently inspected the saccade-locked time course of congruent and incongruent HRs instead of their difference (*Figure 4*, second row). Peaks in enhancement relied on both oscillatory increases in congruent HRs and simultaneous decreases in incongruent HRs. In other words, enhancement peaks appear to reflect a foveal enhancement of target-congruent feature information along with a concurrent suppression of target-incongruent features.

## The influence of saccade latency on foveal congruency effects

On an individual-trial level, the latency of the eye movement may impact foveal detection judgments beyond influencing probe timing. Since the saccade target was visible throughout saccade preparation, peripheral orientation information could accumulate for a longer period of time for long-latency saccades and may have consequently exerted a larger influence on foveal detection. On the other hand, trials with short saccade latencies likely reflect instances in which the saccade target was localized with minimal spatial uncertainty and without erroneous attentional allocation to the opposite hemifield. If short saccade latencies rely on effective target selection and oculomotor planning (see also *Jonikaitis and Theeuwes, 2013*; *Jonikaitis et al., 2017*; *Yan et al., 2018*), congruency effects may emerge most robustly on these trials. To inspect the influence of saccade latency on foveal HRs and FARs, we determined each observer's median saccade latency per target opacity. We then separated all trials into short-latency (md = 247 ± 18 ms) and long-latency (md = 308 ± 18 ms) subsets, depending on whether the latency of the executed saccade on a given trial was shorter or longer than an observer's median latency for the respective opacity level.

All previously described effects were amplified in the subset of short-latency saccades (mean n=1623 trials per observer): across all time points, the difference between congruent and incongruent HRs reached significance for target opacities ≥ 39% and increased with opacity (slope = 2.85% per opacity step, p<0.001; *Figure 5A*). The difference in FARs, in turn, did not vary significantly with opacity (slope = 0.64% per opacity step, p=0.883). Again, we observed a moderately negative yet nonsignificant correlation of r=−0.49 (p=0.181) between the normalized differences in HRs and FARs.

Long-latency saccades, in contrast, showed a markedly different pattern (mean n=1218 trials per observer): enhancement in HRs was overall less pronounced and even decreased with increasing target SNR (slope = −1.23% per SNR step, p=0.044; *Figure 5B*). The differences in HRs and FARs were entirely uncorrelated (r=0.09, p=0.823). These findings suggest that short-latency saccades, which were enabled by effective attentional target selection and oculomotor planning, had driven the enhancement effects reported for the entire trial pool (see *Figure 2*). The small but significant decrease in enhancement with increasing target opacity in the long-latency subset could suggest that long-latency saccades to high-contrast targets in particular rely on attentional lapses, which affected pre-saccadic perceptual processes as well as oculomotor behavior.

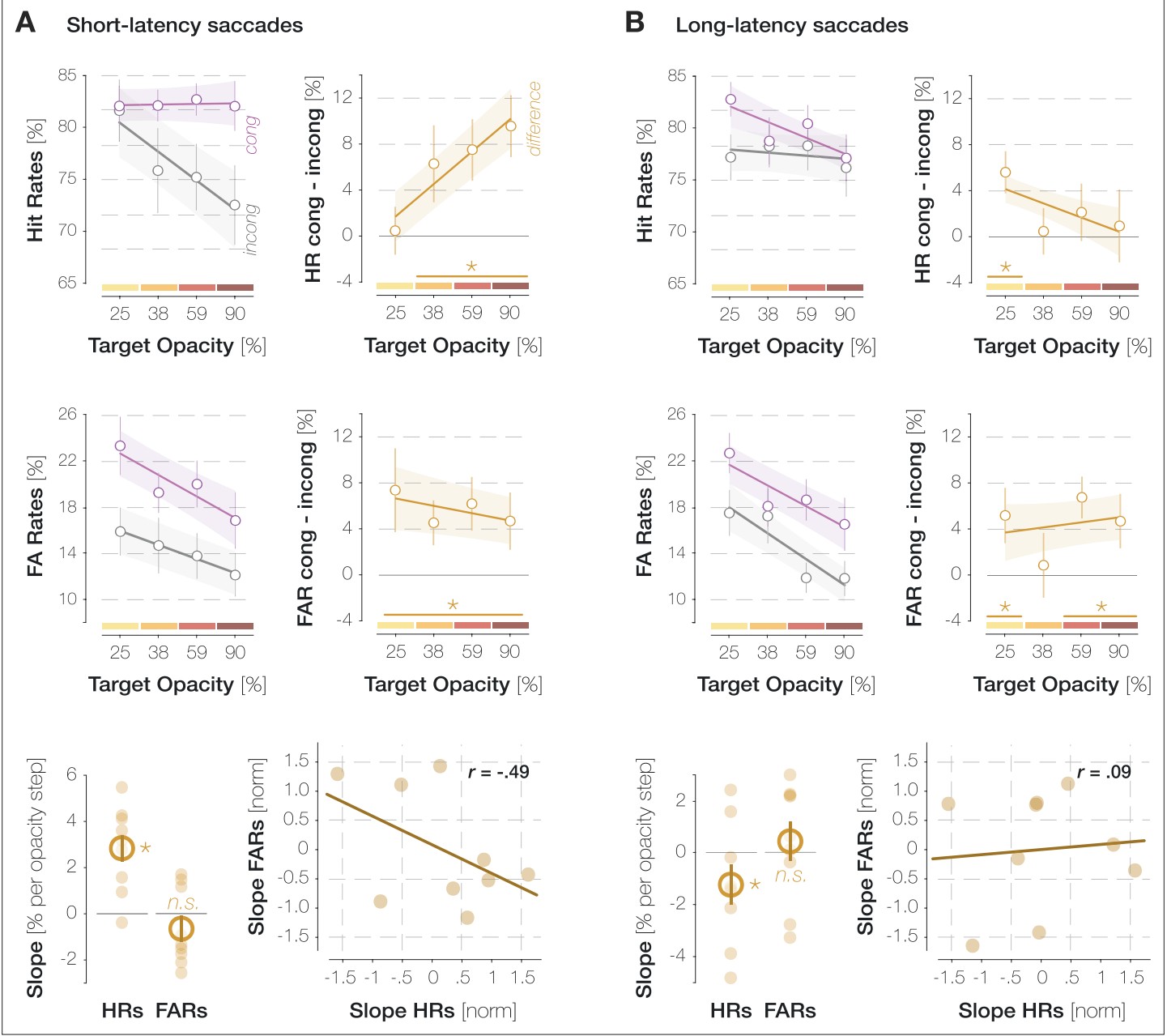

**Figure 5.** Hit rates (HRs) and false alarm rates (FARs) separately for short-latency (**A**) and long-latency (**B**) saccades. All conventions are as in *Figure 2*.

## Discussion

We investigated whether pre-saccadic foveal congruency effects demonstrated in a previous investigation (***Kroell and Rolfs, 2022***) are influenced by the conspicuity of the eye movement target stimulus. For this purpose, we varied the opacity of the orientation-filtered saccade target against the unfiltered 1/f background noise. Along with the SNR of target-like orientation information, this manipulation systematically altered the luminance contrast of the target patch (see Methods). Four observations can be highlighted.

First, the manipulation of target opacity influenced saccade metrics: saccade latencies, as well as saccadic peak velocities, decreased with increasing opacity. The decrease in latencies was to be expected and likely reflects a facilitation of movement planning toward high-contrast target stimuli (***Ludwig et al., 2004***). The small yet significant decrease in peak velocities, which cannot be explained by a concomitant decrease in saccade amplitudes, was unexpected and, to our knowledge, has not

been observed previously. Hypothetically, higher peak velocities may serve to compensate for longer saccade latencies by slightly decreasing the overall time interval between target appearance and saccade landing.

Second, HRs for target-congruent *and* -incongruent foveal probes decreased as target opacity increased, likely because attention was increasingly drawn to the target the more salient it became. Crucially, this decrease was less pronounced for congruent than for incongruent HRs, resulting in a continuous increase of enhancement with target opacity. From an experimenter's perspective, presenting the target at a high opacity (59% and above in our design) seems beneficial since congruency effects, especially when time-resolved, are more robustly detectable. On a more conceptual level, pre-saccadic attention seems to select the target orientation in an automatic fashion, even when the eye movement could easily be programmed based on local contrast variations alone. The influence of peripheral orientation information on foveal detection judgments is directly proportional to the strength of the orientation signal at the target location. As during spontaneous eye movement behavior, saccades are routinely directed toward the currently most conspicuous object in the scene (*'t Hart et al., 2013*). This finding underscores the feasibility of predictive foveal processing in natural visual environments. Of note, the relation between target opacity and foveal enhancement was exclusively driven by saccades with latencies below an observer's median latency in each opacity condition. While the individual-trial relationship between saccade latency and the effect size of pre- and transsaccadic mechanisms is seldom explored (but see *Jonikaitis and Deubel, 2011*; *Jonikaitis and Theeuwes, 2013*; *Jonikaitis et al., 2017*; *Yan et al., 2018*), we suggest that short saccade latencies reflect effective target selection and oculomotor planning, which in turn benefits pre-saccadic perceptual processes such as foveal prediction.

Third, target opacity influenced the time course of enhancement in HRs. When systematic variations of saccade latency were taken into account (*Figure 3B*, third row), congruent and incongruent HRs differed earlier for higher target opacities. These findings suggest that the feedforward processing of the peripheral saccade target may have been accelerated when it was presented at high contrast (*Albrecht et al., 2002*). Alternatively, or in addition, it is conceivable that weaker feedforward signals require a longer accumulation interval before the feedback process can be initiated. Moreover, foveal congruency effects appear to exhibit an oscillatory pattern, with peaks in a medium saccade preparation stage (~80 ms before the eye movement) and immediately before saccade onset. We have noticed this pattern in several investigations with substantially different visual stimuli and behavioral readouts. For instance, using a full-screen dot motion paradigm, we observed a pre-saccadic, small-gain ocular following response to coherent motion in the saccade target region (*Kroell and Rolfs, 2022*, conference abstract; *Kroell and Rolfs, 2022*, dissertation). Predictive ocular following first reached significance ~125 ms before the eye movement, then decreased and subsequently ramped up again ~25 ms before saccade onset. Several explanatory mechanisms appear conceivable. Unlike rapid feedforward processing, feedback propagation has been shown to follow an oscillatory rhythm in the alpha and beta range, i.e., between 7 and 30 Hz (*Bastos et al., 2015*; *Jensen et al., 2015*; *van Kerkoerle et al., 2014*). In our case, it is possible that the object-processing areas that send feedback to retinotopic visual cortex do so at a temporal frequency of ~12.5 Hz. At higher stimulus contrasts, feedforward signals may be fed back instantaneously and without the need for signal accumulation in feedback-generating areas. The resulting perceptual time courses may reflect innate temporal feedback properties most veridically. Alternatively, the initial enhancement peak may be related to the sudden onset of the saccade target stimulus and not to movement preparation itself. In this case, the initial peak should become particularly apparent if enhancement is aligned to the onset of the target stimulus. Yet, *Figures 3 and 4* suggest more prominent oscillations in saccade-locked time courses. In accordance with this, perceptual and attentional processes have been shown to exhibit oscillatory modulations that are phase-locked to action onset (e.g. *Tomassini et al., 2015*; *Hogendoorn, 2016*; *Wutz et al., 2016*; *Benedetto and Morrone, 2017*; *Tomassini et al., 2017*; *Benedetto et al., 2020*). Whether the oscillatory pattern of foveal enhancement, as well as its increased prominence at higher target contrasts, relies on innate temporal properties of feedback signaling, signal accumulation, saccade-locked oscillatory modulations of feedforward processing or attention, or a combination of these factors, one conclusion remains: task-induced cognitive influences suggested to underlie the considerable variability in temporal characteristics of foveal feedback during passive fixation (e.g. *Fan et al., 2016*; *Weldon et al., 2016*; *Weldon et al., 2020*) are not the only possible explanation.

Low-level target properties such as its luminance contrast modulate the resulting time course and should be equally considered, at least in our paradigm.

Fourth, unlike the difference between congruent and incongruent HRs, the difference in FARs did not increase with target opacity. Nonetheless, we suggest that the contribution of different response types to the pool of FAs varied across opacities: as opacity (and, therefore, the difference in HRs) increased, incidental visual properties in the foveal background noise more reliably reflected the subsequently reported orientation. In other words, FAs were increasingly triggered by signal, i.e., orientation information presented on screen, as target opacity increased. Moreover, observers who showed more enhancement in HRs as target opacity increased tended to show a smaller difference between congruent and incongruent FARs with increasing opacity. Combined with a range of arguments provided in our previous publication (*Kroell and Rolfs, 2022*), this observation suggests that foveal congruency effects reflect a variation of pre-saccadic foveal sensitivity rather than a shift in a post-perceptual decision criterion. As noted in the Results section, however, this finding is based on a small number of observations and should be confirmed in larger samples.

## Foveal prediction in natural visual environments

As noted above, human observers typically move their eyes toward the most conspicuous objects in their environment. Foveal prediction seems to benefit from this strategy as the strength of the predicted signal increases with the conspicuity of the eye movement target. Nonetheless, natural visual environments, as well as natural viewing behavior, pose several challenges for the foveal prediction mechanism (see *Kroell and Rolfs, 2022*, for an initial discussion).

First, naturalistic saccade target stimuli will likely exhibit complex shapes and, more often than not, will include feature conjunctions rather than isolated features. Previous findings suggest that the foveal feedback mechanism is capable of operating at this level of complexity: high-level peripheral information such as the category of novel, rendered objects has been successfully decoded from activation in foveal retinotopic cortex (e.g. *Williams et al., 2008*). If, indeed, temporal object-specific areas such as area TE send feedback, the foveal prediction mechanism may even be specialized for the transfer of complex visual properties.

Second, foveal input will often be of high contrast in natural visual environments. Whether fed-back predictive signals can influence foveal perception in the presence of high-contrast feedforward input remains to be established. In our main investigation (*Kroell and Rolfs, 2022*; *Figure 2B*), as well as in previous studies (*Hanning and Deubel, 2022*), pre-saccadic foveal detection performances decreased markedly in the course of saccade preparation, presumably because visuospatial attention gradually shifted toward the saccade target and away from the foveal location. This pre-saccadic decrease in foveal sensitivity may boost the relative weight of fed-back signals by attenuating the conspicuity of high-contrast feedforward input. In other words, the strength of feedforward input to the fovea is reduced gradually across saccade preparation. At the same time, the strength of the fed-back predictive signal should profit from the high contrast of naturalistic saccade targets.

Third, while foveal and peripheral information was congruent on 50% of all 'probe present' trials in our investigation, peripheral and foveal features will often be weakly correlated (see *Samonds et al., 2018*) or even uncorrelated in natural environments. Again, the pre-saccadic attenuation of foveal feedforward processing may allow fed-back peripheral signals to influence perception even if they are uncorrelated with foveal information. Moreover, in piloting variations of our paradigm, we observed that the subjective impression of perceiving the saccade target at the pre-saccadic foveal location is most pronounced if the foveal noise region is replaced with a black Gaussian blob at certain time points before saccade onset (observation by author LMK, unpublished). In consequence, fed-back signals do not seem to require correlated feedforward input to influence perception. Quantitative evidence, however, remains to be established.

Lastly, pre-saccadic foveal input is likely less relevant during natural viewing behavior than it is in our task. It is possible that this task-induced prioritization of the foveal location facilitated the emergence of congruency effects. In a previous experiment (*Kroell and Rolfs, 2022*; *Figure 1D*), however, the perceptual probe could appear anywhere on a horizontal axis of 9 dva length around the screen center. Despite this spatial unpredictability, however, congruency effects peaked at the pre-saccadic foveal location, even after peripheral baseline performances had been raised to a foveal level through an adaptive increase in probe opacity. Ultimately, an influence of task demands on visual processing

can only be fully excluded through techniques that provide a direct readout of perceptual contents without requiring keyboard responses. In psychophysical investigations, a prediction of saccade target motion may be read out from observers' eye velocities (*Kroell and Rolfs, 2022*; *Kwon et al., 2019*). In electroencephalographic and neurophysiological studies, foveal predictions should manifest in early visually evoked potentials (e.g. *Creel, 2019*) and increased firing rates of feature-selective foveal neurons in early visual areas, respectively.

## Neural implementation of foveal prediction

Based on the body of our findings as well as previous literature, we suggested a parsimonious feedback mechanism to underlie the observed effects: the preparation of a saccadic eye movement, and the concomitant shift of pre-saccadic attention (e.g. *Kowler et al., 1995*; *Deubel, and Schneider, 1996*), selects the peripheral target stimulus among competing information. Higher-order visual areas feed selected feature input back to early retinotopic areas—specifically, to neurons with foveal receptive fields. Fed-back feature information combines with congruent, foveal feedforward input, resulting in the enhancement effects we observe. Especially in the context of active vision, this feedback mechanism is appealing as it resolves a combinatorial issue associated with feature-specific information transfer before saccades. Consider a simplified case in which, right before a saccadic eye movement, the activation of a feature-selective neuron that encodes a certain retinal location is transferred to a neuron within the same brain area that will encode said retinal location after saccade landing. For this mechanism to function for any possible saccade direction and amplitude, most neurons would need to be connected to most other neurons (or, in a simplified version, to neurons with foveal receptive fields) in a given brain area. Assuming an information transmission via feedback rather than horizontal connections significantly reduces this dimensionality: higher-order visual areas that encode object properties (largely) detached from retinotopic or spatiotopic reference frames selectively transfer feature information to neurons with foveal receptive fields, irrespective of the vector of the upcoming saccade. This parsimonious mechanism would have shortcomings. In particular, foveal feedback should become less effective during saccade sequences where several peripheral targets are simultaneously attended. Feature information at both attended target locations may be fed back in temporal succession or weighted and erroneously combined into a single fed-back signal. In most cases, however, foveal feedback may reasonably achieve what established transsaccadic mechanisms struggle to explain: an anticipation of the features of a single saccade target—which typically constitutes the currently most relevant object in the visual field—in foveal vision.

While direct feedback connections from higher-order to early visual areas would constitute the most straightforward implementation, it is conceivable that feedback signals are relayed through and modulated by subcortical areas. In particular, the thalamic pulvinar has been identified as a connection hub for visual processing that receives copies of feedforward and feedback connections from different visual areas and may even combine information across visual space (*Cortes et al., 2024*). In the case of foveal prediction, thalamic neurons may receive fed-back signals from higher-order areas and enhance those signals before passing them on to cortical neurons with foveal receptive fields. Perhaps a modification of foveal activation within the thalamic pulvinar itself is sufficient to influence perception. To the best of our understanding, however, the fed-back signal must originate in non-retinotopic, higher-order object processing areas to reduce the number of necessary neuronal connections.

## Conclusion

To sum up, this investigation both provides further mechanistic insights into the pre-saccadic foveal prediction mechanism and constrains the parameter space for researchers planning to adapt our paradigm in future studies. Conceptually, active peripheral feature sampling is not necessary for foveal prediction to emerge. Instead, foveal congruency effects develop even when (or especially when) salient local contrast variations at the saccade target location can be used to direct the eye movement. In consequence, presenting the target at a high opacity appears purely beneficial. At high target opacities, foveal congruency effects in HRs are more pronounced. Moreover, observers respond more systematically to target-like orientations in foveally presented noise, facilitating reverse-correlation-based analyses.

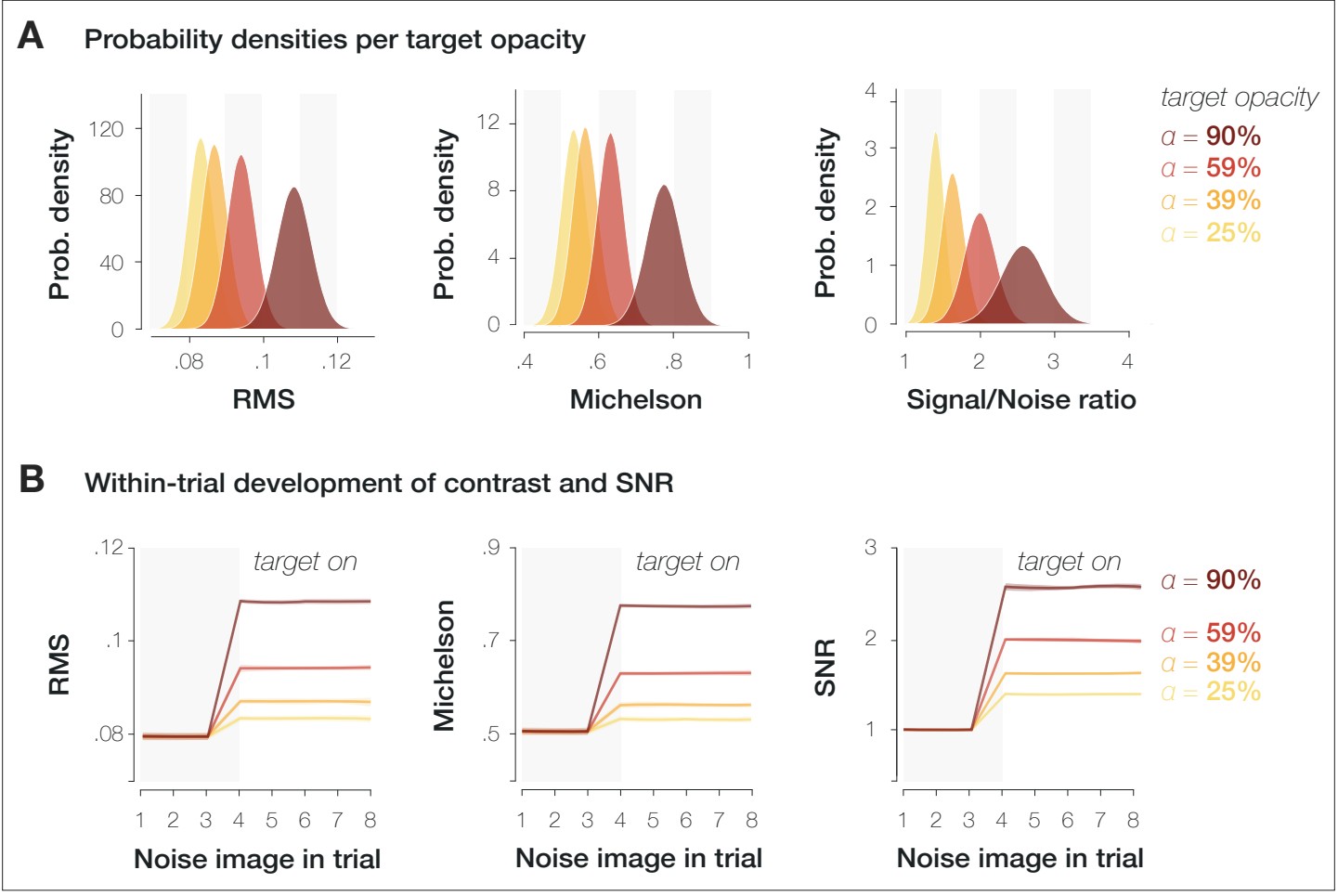

**Figure 6.** Saccade target properties at different opacity levels. Variation of root mean square (RMS) contrast (first column), Michelson contrast (second column), and signal-to-noise ratio (SNR) (third column) within the saccade target region. (**A**) Probability density distributions per measure and target opacity (yellow to dark red shadings). (**B**) Mean and standard deviation of contrast and SNR separately for each noise image presented during the saccade preparation period. The target was presented from the fourth noise image on.

## Methods

### Sample

Nine human observers (six females, eight right-handed, six right-eye dominant, no authors) aged 22–30 years (mdn = 26.0) participated in the experiment. Normal (n=3) or corrected-to-normal (n=6) visual acuity was ensured at the beginning of the first session using a Snellen chart (*Hetherington, 1954*) embedded in a Polatest vision testing instrument (Zeiss, Oberkochen, Germany). Observers yielding scores of 20/25 or 20/20 were invited to proceed with the experiment. Ocular dominance was assessed using the Miles test (*Miles, 1930*). Since data collection was performed during the COVID-19 pandemic, our sample was composed of lab members. Nonetheless, all observers were naïve as to the purpose of the study. Participants gave written informed consent before the experiments and were compensated with either accreditation of work hours or a payment of 8.50€/hour plus a bonus of 1€/ session. The study complied with the Declaration of Helsinki (2013) and was approved by the Ethics Committee of the Department of Psychology at Humboldt-Universität zu Berlin. The research question, experimental paradigm, and data analyses were preregistered on the Open Science Framework (https://osf.io/wceba). Preprocessed data and experimental code are available here: https://osf.io/ up5ck/files/osfstorage.

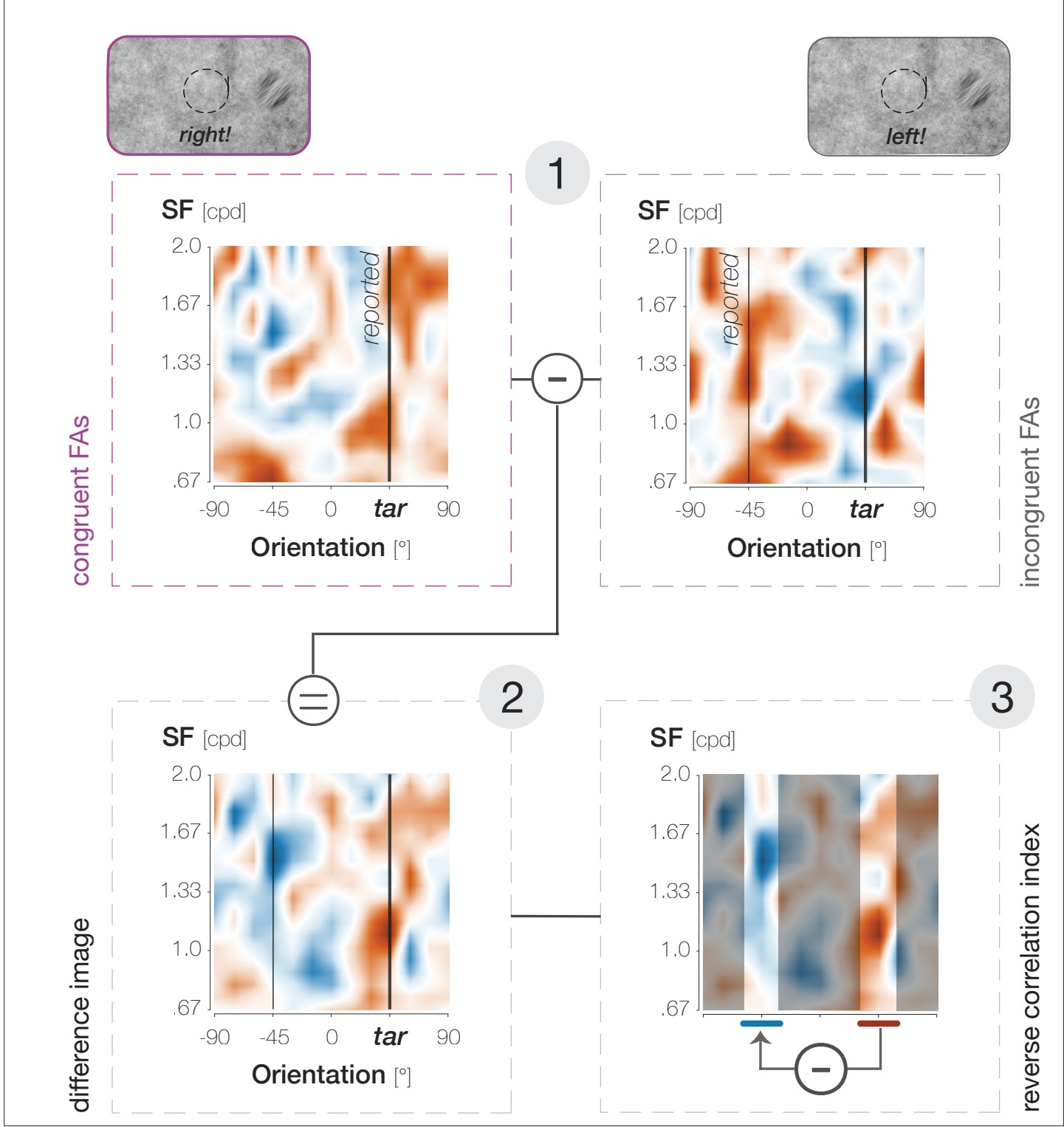

**Figure 7.** Calculation of the reverse correlation index plotted in *Figure 2C*, illustrated for the 59% target opacity condition. In step 1, we identified the average properties (SF*orientation) of the foveal noise window on congruent (purple outlines and font) and incongruent (gray outlines and font) FA trials. In step 2, we determined the difference between them (congruent-incongruent). In step 3, we subtracted filter energies around the non-target orientation (–45°) from filter energies around the target orientation (45°).

## Experimental setup and design

The external setup, session structure, and trial time course were identical to our previous experiments (*Kroell and Rolfs, 2022*) with the following exceptions: first and most importantly, the saccade target was overlaid on the background noise with one of four logarithmically spaced opacities (α=25%, 39%, 59%, and 90% compared to 60% in *Kroell and Rolfs, 2022*). Targets with different opacities were presented in separate experimental blocks, the order of which was randomized. Before every block, we showed a preview of the target stimulus at the respective opacity (sequentially at both possible target locations in the left and right screen half, randomly oriented either 45° to the left or right). Second, during each trial in a block, we removed the target upon saccade initiation, i.e., once gaze had crossed a boundary of 2.0 dva diameter around the fixation dot. Third, we increased the number of potential background noise images from 17 to 27 images per session and observer. All 27 images were generated at the beginning of a given session. Before each trial, we randomly drew 16 images from this pool (without replacement) and presented them in random order. Since 9 observers each completed 7 sessions, 27*9*7=1701 different noise images were presented in the course of the experiment. Lastly, observers completed four staircase blocks before the main experiment in every session (one block per target opacity). Individual staircase blocks were short (32 trials each) and served to familiarize observers with all target opacities, as well as to obtain an estimate of observers' typical saccade latency for each opacity. Saccade latency estimates were used to define three possible probe onset times for each target opacity in the main experiment. Importantly, however, the same staircase ran throughout all four blocks and returned one optimal probe opacity estimate that was used throughout the main experiment, irrespective of target opacity.

Increasing target opacity increased the luminance contrast and the SNR of the filtered orientation within the saccade target region. During the experiment, we manipulated target opacity by varying the 'globalAlpha' input to the Psychtoolbox (*Kleiner et al., 2007*) function 'DrawTexture' and enabled alpha blending with the inputs 'GL_SRC_ALPHA' (source) and 'GL_ONE_MINUS_SRC_ALPHA' (destination). To emulate the functionality of this presentation technique, we determined the luminance sum of the background noise and the orientation-filtered target patch multiplied with the respective opacity:

$$luminanceCombined = luminanceBackground + opacity * (luminanceFiltered - luminanceBackground) * cosineMask$$

where 'luminanceCombined' denotes the resulting luminance values within the target region, 'luminanceBackground' denotes the luminance values within a square of 3 dva width around the center of the target location, 'opacity' denotes the manipulated target opacity (25–90%), 'luminanceFiltered' denotes the luminance values of the orientation-filtered target patch (3 dva width), and 'cosineMask' denotes the raised cosine mask with which the filtered patch was overlayed (3 dva diameter). We then calculated the root mean square (RMS, see *Peli, 1990*) and *Michelson, 1927* of the resulting luminance sum for every trial. To compute the SNR of the target orientation, we determined the energy of various SF and orientation combinations in the combined luminance patch (see Data analysis). We then divided the mean filter energy within a region of ±15° around the target orientation by the mean filter energy for all remaining orientations. An increase in target opacity from 25% to 90% corresponded to an increase in RMS contrast from 0.08 to 0.11, an increase in Michelson contrast from 0.53 to 0.78, and an increase in the SNR of the filtered orientation from 1.40 to 2.58 (*Figure 6*).

## Data analysis

Online trial abortion criteria, as well as trial exclusions based on offline gaze analyses, are identical to our previous study (*Kroell and Rolfs, 2022*). Unlike in our previous investigation, we intended to remove the saccade target during saccadic flight on all trials. Across observers, 0.24% (min = 0%; max = 1.1%; std = 0.40%) of trials were excluded due to the saccade target still being visible after saccade landing.

To test the influence of target opacity on the difference between congruent and incongruent HRs and FARs, we performed linear mixed-effects models in which we described the variance in both differences ($HR_{cong}$–$HR_{incong}$; $FAR_{cong}$–$FAR_{incong}$) across all time points with a fixed effect of target opacity and a random intercept for observer:

$$\text{cong} - \text{incong opacity} + (1\,|\,\text{observer})$$

where 'cong-incong' refers to the difference between either congruent and incongruent HRs or congruent and incongruent FARs. As stated in the Results section, this model outperformed the simplest model including a fixed effect of target opacity only (*Albrecht et al., 2002*) and a more complex one involving a random intercept and random slope for observer (*Bastos et al., 2015*):

$$(1)\,\text{cong} - \text{incong} \sim \text{opacity}$$

$$(2)\,\text{cong} - \text{incong} \sim \text{opacity} + (\text{opacity}\,|\,\text{observer})$$

Model fitting was performed with the MATLAB function fitlme (MATLAB 2020b, Mathworks, Natick, MA, USA).

All remaining tests of statistical significance relied on bootstrapping: within each observer, we determined the means in the to-be-compared conditions and computed the difference between those means. Across observers, we drew 10,000 random samples from these differences (with replacement). Reported p-values correspond to the proportion of differences smaller than or equal to zero. We considered p-values≤0.05 significant.

Furthermore, to relate the difference between congruent and incongruent HRs to the difference in FARs, we z-standardized HR and FAR slopes before computing the Pearson's correlation coefficient.

Lastly, we used a three-step approach to calculate the reverse correlation index plotted in *Figure 2C* (see *Figure 7*). First, we determined all noise images that had been visible during saccade preparation on trials in which observers generated a congruent or incongruent FA. We then described the average visual properties of these noise images within a 3 dva diameter circular region around the screen center using a set of Gabor filters with varying SF*orientation properties (SFs from 0.67 to 2.0 cycles per degree [cpd] in 20 equal steps of 0.07 cpd; orientations from –90° to 90° in 13 equal steps of 15°; see *Movellan, 2002*; *Wyart et al., 2012*; *Li et al., 2016*; *Schweitzer and Rolfs, 2020*; *Schweitzer and Rolfs, 2021*; details provided in *Kroell and Rolfs, 2022*). Second, we subtracted the filter energies on incongruent FA trials from filter energies on congruent FA trials on an individual-observer level. Third, we selected a 30° orientation window around the non-target orientation (i.e. –45°) and subtracted the mean filter energy in that window from the mean filter energy in a 30° orientation window around the target orientation (i.e. 45°). The higher the resulting value, the more congruent and incongruent FAs were based on signal, i.e., on incidental orientation information on screen. Note that we combined the lower two and the higher two target opacities to increase the number of trials and obtain reliable reverse correlation measures. The resulting index is based on m=2461 (std = 583) and m=1952 (std = 629) trials per observer for the lower two and higher two opacities, respectively.

## Additional information

### Funding

| Funder | Grant reference number | Author |
| --- | --- | --- |
| Deutsche Forschungsgemeinschaft | RO3579/8-1 | Martin Rolfs |
| Deutsche Forschungsgemeinschaft | RO3579/9-1 | Martin Rolfs |
| Deutsche Forschungsgemeinschaft | RO3579/12-1 | Martin Rolfs |

The funders had no role in study design, data collection and interpretation, or the decision to submit the work for publication.

### Author contributions

Lisa M Kroell, Conceptualization, Data curation, Formal analysis, Investigation, Visualization, Methodology, Writing – original draft, Project administration; Martin Rolfs, Conceptualization, Resources, Supervision, Funding acquisition, Project administration, Writing - review and editing

## Author ORCIDs
Lisa M Kroell ⓘ https://orcid.org/0000-0002-3508-5214
Martin Rolfs ⓘ https://orcid.org/0000-0002-8214-8556

## Ethics

The study complied with the Declaration of Helsinki (2013) and was approved by the Ethics Committee of the Department of Psychology at Humboldt-Universität zu Berlin (internal approval number: 2019-08). All participants provided written informed consent prior to participation and were informed of their right to withdraw from the experiment at any time. Compensation was provided either in the form of accredited work hours or monetary payment (€8.50 per hour plus a €1 bonus per session).

Reviewer #1 (Public review): https://doi.org/10.7554/eLife.91236.4.sa1
Reviewer #3 (Public review): https://doi.org/10.7554/eLife.91236.4.sa2
Author response https://doi.org/10.7554/eLife.91236.4.sa3

# Additional files

## Supplementary files
MDAR checklist

## Data availability

Trial variables for all subjects and sessions, preprocessed saccade characteristics as well as the experimental code are available on the Open Science Framework: https://osf.io/up5ck/. We uploaded the noise images and design files for all seven sessions of subject 1 ('CB') as an example (see folder 'noiseImages_Design').

The following dataset was generated:

| Author(s) | Year | Dataset title | Dataset URL | Database and Identifier |
|---|---|---|---|---|
| Kroell LM, Rolfs M | 2025 | Data for: The magnitude and time course of pre-saccadic foveal prediction depend on the conspicuity of the saccade target (eLife VOR) | https://osf.io/up5ck/ | Open Science Framework, up5ck |

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

## Appendix 1

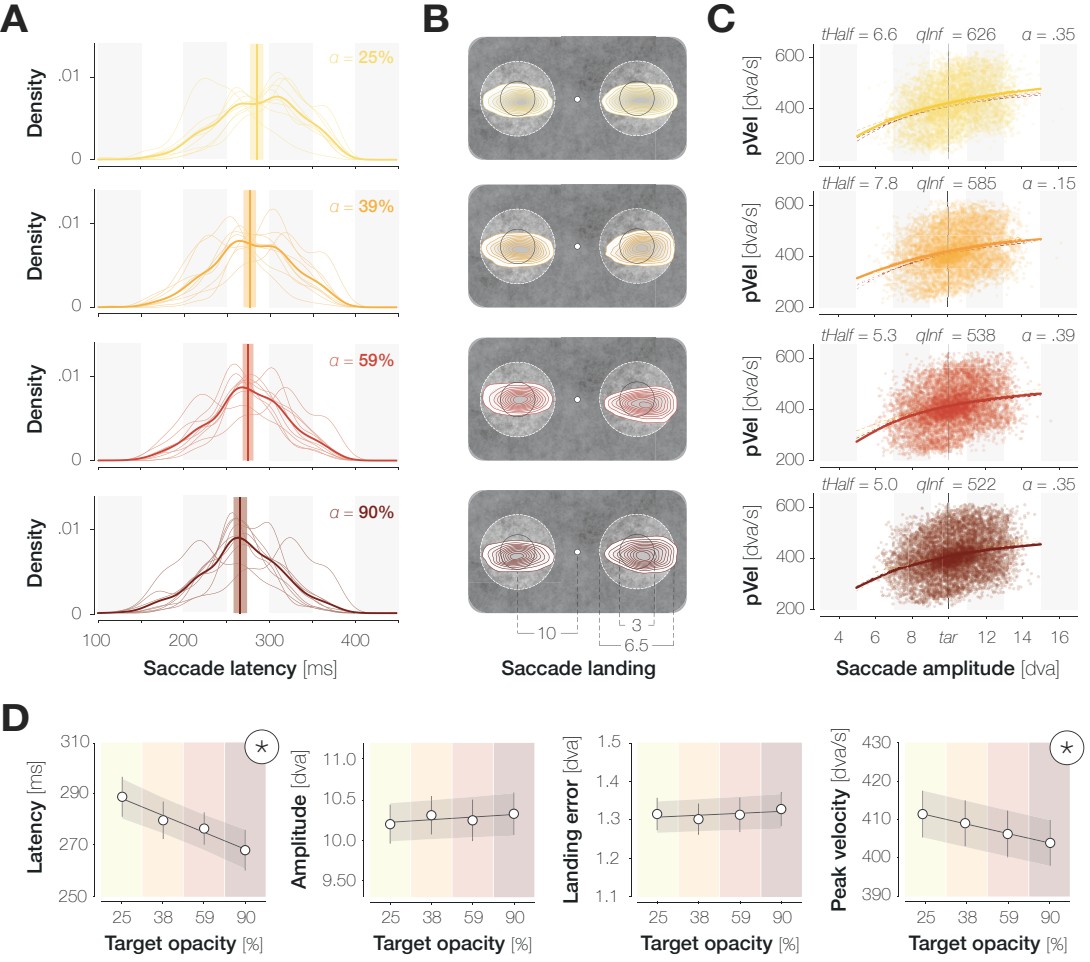

**Appendix 1—figure 1.** The influence of target opacity on saccade metrics. (**A**) Probability density distributions of saccade latencies for different, increasing target opacities (from top to bottom; see **Figure 3A**). Distributions with thin and thick lines represent individual-observer and mean probability densities, respectively. Vertical lines and shaded regions represent median latencies and standard errors. (**B**) Bivariate Gaussian kernel densities of saccade landing coordinates separately for leftward and rightward saccades. The distance between the fixation and target locations was reduced for illustration purposes (see legend). (**C**) Main sequences defined as the relation between saccade amplitudes and peak velocities. Dots symbolize individual trials (n~29,000). Fitted lines represent the average of logistic function fits to individual-observer data (**Conder, 2023**). The mean parameters of each fit are provided above the respective panel ('tHalf': symmetric inflection point; 'qInf': horizontal asymptote; α: decay constant). (**D**) Summary plots for saccade latency, amplitude, landing error, and peak velocity. Dots represent median (latency) and mean (amplitude, error, velocity) values across observers, and error bars represent the respective standard errors. Black lines and shaded error bands represent the mean of linear fits to individual-observer data and their standard errors, respectively. Asterisks highlight slopes that are significantly different from zero (determined via bootstrapping, n=9 observers, p<0.05).

