## [Editor Report · eLife Assessment]

This study reports **important** findings about pre-saccadic foveal prediction and the extent to which it is influenced by the visibility of the saccade target relative to its background. The research methodology and results make a **convincing** case that foveal congruency effects develop when salient local contrast variations at the saccade target location can be used to direct the eye movement. This work should be of broad interest to visual neuroscientists, as well as those interested in understanding perception in the context of eye movements and in modeling visually guided actions.

---

## [Referee Report · Reviewer #1 (Public review)]

Summary:

This study provides new insights on the phenomenon of pre-saccadic foveal prediction previously reported by the same authors. In particular, this study examines to what extent this phenomenon varies based on the visibility of the saccade target. Visibility is defined as the contrast level of the target with respect to the noise background, and it is related to the signal-to-noise ratio of the target. A more visible target facilitates the oculomotor behavior planning and execution, however, as speculated by the authors, it can also benefit foveal prediction even if the foveal stimulus visibility is maintained constant. Remarkably, the authors show that presenting a highly visible saccade target is beneficial for foveal vision as detection of stimuli with an orientation similar to that of the saccade target is improved, the lower is the saccade target visibility, the less prominent is this effect. The results are convincing and the research methodology is technically sound.

Comments on revisions:

The authors addressed all the concerns raised in the previous rounds of reviews.

---

## [Referee Report · Reviewer #3 (Public review)]

Summary:

In this manuscript, the authors ran a dual task. Subjects monitored a peripheral location for a target onset (to generate a saccade to), and they also monitored a foveal location for a foveal probe. The foveal probe could be congruent or incongruent with the orientation of the peripheral target. In this study, the authors manipulated the conspicuity of the peripheral target, and they saw changes in performance in the foveal task.

Comments on revisions:

The authors have addressed all comments. Thanks.

---

## [Author Response]

The following is the authors’ response to the previous reviews

**Reviewer #1 (Public review):**
Summary:This study examines to what extent this phenomenon varies based on the visibility of the saccade target. Visibility is defined as the contrast level of the target with respect to the noise background, and it is related to the signal-to-noise ratio of the target. A more visible target facilitates the oculomotor behavior planning and execution, however, as speculated by the authors, it can also benefit foveal prediction even if the foveal stimulus visibility is maintained constant. Remarkably, the authors show that presenting a highly visible saccade target is beneficial for foveal vision as detection of stimuli with an orientation similar to that of the saccade target is improved, the lower is the saccade target visibility, the less prominent is this effect.Strengths:The results are convincing and the research methodology is technically sound.Weaknesses:It is still unclear why the pre-saccadic enhancement would oscillate for targets with higher opacity levels, and what would be the benefit of this oscillatory pattern. The authors do not speculate too much on this and loosely relate it to feedback processes, which are characterized by neural oscillations in a similar range.

We thank the reviewer for their assessment. We intentionally decided to describe the oscillatory pattern without claiming to be able to pinpoint its origin. The finding was incidental and, based on psychophysical data alone, we would not feel comfortable doing anything but loosely relating it to potential mechanisms on an explicitly speculative basis. In the potential explanation we provide in the manuscript, the oscillatory pattern would likely not serve a benefit–rather, it would constitute an innate consequence and, thus, a coincidental perceptual signature of potential feedback processes.

**Reviewer #2 (Public review):**
Summary:In this manuscript, the authors ran a dual task. Subjects monitored a peripheral location for a target onset (to generate a saccade to), and they also monitored a foveal location for a foveal probe. The foveal probe could be congruent or incongruent with the orientation of the peripheral target. In this study, the authors manipulated the conspicuity of the peripheral target, and they saw changes in performance in the foveal task. However, the changes were somewhat counterintuitive.

We regret that our findings remain counterintuitive to the reviewer even after our extensive explanations in the previous revision round and the corresponding changes in the manuscript. We repeat that both the decrease in foveal Hit Rates and the increase in foveal enhancement with increasing target contrast were expected and preregistered prior to data collection.

Strengths:The authors use solid analysis methods and careful experimental design.Comments on revisions:The authors have addressed my previous comments.One minor thing is that I am confused by their assertion that there was no smoothing in the manuscript (other than the newly added time course analysis). Figure 3A and Figure 6 seem to have smoothing to me.

When the reviewer suggested that the “data appear too excessively smoothed” in the first revision, we assumed that they were referring to pre-saccadic foveal Hit and False Alarm rates, not to fitted distributions. As we state in the legend of Figure 3A (as well as in Figures 6 and S1), the “smoothed” curves constitute the probability density distributions of our raw data. Concerning the energy maps resulting from reverse correlation analyses, we described our proceeding in detail in our initial article (Kroell & Rolfs, 2022):

“Using this method, we obtained filter responses for 260 SF*ori combinations per noise image (Figure 6 in Materials and methods, ‘Stimulus analysis’). SFs ranged from 0.33 to 1.39 cpd (in 20 equal increments). Orientations ranged from –90–90° (in 13 equal increments). To normalize the resulting energy maps, we z-transformed filter responses using the mean and standard deviation of filter responses from the set of images presented in a certain session. To obtain more fine-grained maps, we applied 2D linear interpolations by iteratively halving the interval between adjacent values 4 times in each dimension. To facilitate interpretability, we flipped the energy maps of trials in which the target was oriented to the left. In all analyses and plots,+45° thus corresponds to the target’s orientation while –45° corresponds to the other potential probe orientation. Filter responses for all response types are provided at https://osf.io/v9gsq/.”

We have added a pointer to this explanation to the current manuscript (see line 836).

Another minor comment is related to the comment of Reviewer 1 about oscillations. Another possible reason for what looks like oscillations is saccadic inhibition. when the foveal probe appears, it can reset the saccade generation process. when aligned to saccade onset, this appears like a characteristic change in different parameters that is time-locked to saccade onset (about a 100 ms earlier). So, maybe the apparent oscillation is a manifestation of such resetting and it's not really an oscillation. so, I agree with Reviewer 1 about removing the oscillation sentence from the abstract.

While we understand that a visible probe will result in saccadic inhibition (White & Rolfs, 2016), we are unsure how a resetting of the saccade generation process should manifest in increased perceptual enhancement of a specific, peripheral target orientation in the presaccadic fovea. Moreover, as we describe in our initial article (Kroell & Rolfs, 2022), we updated the background noise image every 50 ms and embedded our probe stimulus into the surrounding noise using smooth orientation filters and raised cosine masks to avoid a disruptive influence of probe appearance on movement planning and execution (Hanning, Deubel, & Szinte, 2019). And indeed, we demonstrated that the appearance of the foveal probe did not disrupt saccade preparation, that is, did not increase saccade latencies compared to ‘probe absent’ trials in which no foveal probe was presented (see Kroell & Rolfs, 2022; sections “Parameters of included saccades in Experiment 1” and “Parameters of included saccades in Experiment 2”). In the current submission, saccade latencies in ‘probe present’ trials exceeded saccade latencies in ‘probe absent’ trials by a mere 4.7±2.3 ms. Additionally, to inspect the variation of saccade execution frequency directly, we aligned the number of saccade generation instances to the onset of the foveal probe stimulus (see Author response image 1). In line with what we described in a previous paradigm employing flickering bandpass filtered noise patches (Kroell & Rolfs, 2021; 10.1016/j.cortex.2021.02.021), we observed a regular variation in saccade execution frequency that reflected the duration of an individual background noise image (50 ms in this investigation). In other words, the repeated dips in saccadic frequency are likely caused by the flickering background noise and not the onset of the foveal probe which would produce a single dip ~100 ms after probe onset. Given these results, we do not see a straight-forward explanation for how the variation of saccade execution frequency in 20 Hz intervals would boost peripheral-to-foveal feature prediction before the saccade in ~10 Hz intervals. Nonetheless, we removed the sentence referencing oscillations from the Abstract.

**Recommendations for the authors:**

**Reviewer #1 (Recommendations for the authors):**
Overall, The authors did a good job in addressing the points I raised. Two new sections were added to the manuscript, one to address how the mechanisms of foveal predictions would play out in natural viewing conditions, and another one examining more in depth the potential neural mechanisms implicated in foveal predictions. I found these two sections to be quite speculative, and at points, a bit convoluted but could help the reader get the bigger picture. I still do not have a clear sense of why the pre-saccadic enhancement would oscillate for targets with higher opacity levels, and what would be the benefit of this oscillatory pattern. The authors do not speculate too much on this and loosely relate it to feedback processes, which are characterized by neural oscillations in a similar range.

Please see our response to ‘Weaknesses’.

I still find this a loose connection and would suggest removing the following phrase from the abstract "Interestingly, the temporal frequency of these oscillations corresponded to the frequency range typically associated with neural feedback signaling".

We have removed this phrase.

Finally, the authors should specify how much of this oscillation is due to oscillations in HR of cong vs. oscillations in HR of incongruent trials or both.

We fitted separate polynomials to congruent and incongruent Hit Rates instead of their difference. Peaks in enhancement relied on both, oscillatory increases in congruent Hit Rates and simultaneous decreases in incongruent Hit Rates. In other words, enhancement peaks appear to reflect a foveal enhancement of target-congruent feature information along with a concurrent suppression of target-incongruent features. We added this paragraph and Figure 4 to the Results section.

Additional changes:

Two figures had accidentally been labeled as Figure 5 in our first revision. We corrected the figure legends and all corresponding figure references in the text.